# Time-Frequency Characterization of Resting Brain in Bipolar Disorder during Euthymia—A Preliminary Study

**DOI:** 10.3390/brainsci11050599

**Published:** 2021-05-07

**Authors:** Adrian Andrzej Chrobak, Bartosz Bohaterewicz, Anna Maria Sobczak, Magdalena Marszał-Wiśniewska, Anna Tereszko, Anna Krupa, Anna Ceglarek, Magdalena Fafrowicz, Amira Bryll, Tadeusz Marek, Dominika Dudek, Marcin Siwek

**Affiliations:** 1Department of Adult Psychiatry, Jagiellonian University Medical College, Kopernika st. 21a, 31-501 Kraków, Poland; adrian.chrobak@uj.edu.pl (A.A.C.); dominika.dudek@uj.edu.pl (D.D.); 2Department of Psychology of Individual Differences, Psychological Diagnosis and Psychometrics, Faculty of Psychology in Warsaw, SWPS University of Social Sciences and Humanities, Chodakowska st. 19/31, 03-815 Warsaw, Poland; bohaterewicz@gmail.com (B.B.); mmarszal@swps.edu.pl (M.M.-W.); 3Department of Cognitive Neuroscience and Neuroergonomics, Jagiellonian University, Prof. Stanisława Łojasiewicza st. 4, 30-348 Kraków, Poland; aniaewacc@gmail.com (A.C.); vonfrovitz@gmail.com (M.F.); tademarek@gmail.com (T.M.); 4Chair of Psychiatry, Jagiellonian University Medical College, Kopernika st. 21a, 31-501 Kraków, Poland; tereszka@gmail.com; 5Department of Psychiatry, Jagiellonian University Medical College, Kopernika st. 21a, 31-501 Kraków, Poland; annajuliakrupa@doctoral.uj.edu.pl; 6Malopolska Centre of Biotechnology, Neuroimaging Group, Jagiellonian University, Gronostajowa st. 7a, 30-387 Kraków, Poland; 7Department of Radiology, Jagiellonian University Medical College, Kopernika st. 19, 31-501 Kraków, Poland; amira.bryll@uj.edu.pl; 8Department of Affective Disorders, Jagiellonian University Medical College, Kopernika st. 21a, 31-501 Kraków, Poland; marcin.siwek@uj.edu.pl

**Keywords:** ALFF, f/ALFF, ReHo, functional connectivity, insula, affective disorders

## Abstract

The goal of this paper is to investigate the baseline brain activity in euthymic bipolar disorder (BD) patients by comparing it to healthy controls (HC) with the use of a variety of resting state functional magnetic resonance imaging (rs-fMRI) analyses, such as amplitude of low frequency fluctuations (ALFF), fractional ALFF (f/ALFF), ALFF-based functional connectivity (FC), and r egional homogeneity (ReHo). We hypothesize that above-mentioned techniques will differentiate BD from HC indicating dissimilarities between the groups within different brain structures. Forty-two participants divided into two groups of euthymic BD patients (n = 21) and HC (n = 21) underwent rs-fMRI evaluation. Typical band ALFF, slow-4, slow-5, f/ALFF, as well as ReHo indexes were analyzed. Regions with altered ALFF were chosen as ROI for seed-to-voxel analysis of FC. As opposed to HC, BD patients revealed: increased ALFF in left insula; increased slow-5 in left middle temporal pole; increased f/ALFF in left superior frontal gyrus, left superior temporal gyrus, left middle occipital gyrus, right putamen, and bilateral thalamus. There were no significant differences between BD and HC groups in slow-4 band. Compared to HC, the BD group presented higher ReHo values in the left superior medial frontal gyrus and lower ReHo values in the right supplementary motor area. FC analysis revealed significant hyper-connectivity within the BD group between left insula and bilateral middle frontal gyrus, right superior parietal gyrus, right supramarginal gyrus, left inferior parietal gyrus, left cerebellum, and left supplementary motor area. To our best knowledge, this is the first rs-fMRI study combining ReHo, ALFF, f/ALFF, and subdivided frequency bands (slow-4 and slow-5) in euthymic BD patients. ALFF, f/ALFF, slow-5, as well as REHO analysis revealed significant differences between two studied groups. Although results obtained with the above methods enable to identify group-specific brain structures, no overlap between the brain regions was detected. This indicates that combination of foregoing rs-fMRI methods may complement each other, revealing the bigger picture of the complex resting state abnormalities in BD.

## 1. Introduction

Bipolar disorder (BD) is a chronic and debilitating mental illness, associated with the presence of recurrent periods of depression, mania (BD type I), hypomania (BD type II), or mixed episodes. BD is characterized by affective symptoms and cognitive impairments, not only during the acute affective episodes, but also the periods of clinical remission (euthymia) [1]. Advancements in the field of functional magnetic resonance imaging (fMRI) methods enable capturing complex brain activity alterations during task-negative state in BD [2,3]. Resting-state fMRI (rs-fMRI) reflects the neuronal baseline brain activity during the absence of goal-directed external stimuli. Additional strength of the above procedure is the lack of task to perform. Instead, participants are typically asked to rest with their eyes open and not to have meaningful thoughts for several minutes [4]. There is a large number of methods of analysis in rs-fMRI. The most popular one is functional connectivity (FC), which indicates the temporal correlation of the blood oxygenation level dependent (BOLD) signal between functionally-linked structures [5]. The amplitude of low frequency fluctuations (ALFF), on the other hand, detects fluctuations lower than cardiac or respiratory ones, which are considered to be related to spontaneous neural activity within a region [6]. Another variant of ALFF is the fractional amplitude of low frequency fluctuations (f/ALFF), which is computed as the fractional sum of amplitudes within the low-frequency range (0.01–0.08 Hz) that was divided by the sum of amplitude across the entire frequency range [7]. Moreover, a recent approach called “ALFF-based FC analysis” has been proposed, which combines the aforementioned methods. This technique enables evaluation of interaction and association between regional resting-state amplitude of low frequency oscillations and their network-based temporal correlations with other brain regions [8]. Another approach is to look at the respective frequencies of BOLD signal. The arbitrary selection of the frequency band may cause information loss in other ranges, thus using a variety of frequency bands allows to obtain complementary information on spontaneous neural activity [9]. Moreover, previous studies suggested that subdivided frequency bands, including slow-5 (0.01–0.027 Hz) and slow-4 (0.027–0.073 Hz) can contribute to further between-group characterization of the rs-fMRI dataset [10]. Regional homogeneity (ReHo) is an index that enables to identify the similarities or coherence of intraregional spontaneous frequencies [11]. Altered ReHo indicates changes in spontaneous neuronal activity at the local level or atypical patterns of neuronal synchrony across global networks [12].

So far, functional connectivity studies in bipolar patients have revealed the decreased connectivity among the ventral prefrontal cortex and amygdala [13,14]. Previous studies using ALFF analysis, on the other hand, have shown significant differences between bipolar patients and controls, such as increased ALFF in the ventral and dorsolateral prefrontal cortex, frontal eye field, insula, putamen, as well as decreased ALFF in the lingual gyrus [15]. Moreover, ReHo studies have shown altered activity of the left ventral visual stream [12], left medial frontal gyrus, as well as left inferior parietal lobe [16]. Studies of BD patients performed during euthymic state are extremely scarce, nevertheless one study exhibited significant hyper-connectivity between the right amygdala and right ventrolateral prefrontal cortex in patients compared to healthy subjects, which was partially mediated by activity in the anterior cingulate cortex [17]. Another study evaluating both f/ALFF and ALFF in euthymic BD patients with history of psychotic symptoms showed abnormal regional activation in fronto-striatal regions [18]. Whole-brain ReHo has been so far evaluated during euthymia only in the pediatric BD patients group, indicating reduced signal in superior temporal gyrus and superior parietal lobe [19]. According to our knowledge, no rs-fMRI analysis using ALFF, f/ALFF, and subdivided frequency bands (slow-4 and slow-5) in the euthymic BD patients has been conducted yet. The goal of this paper is to investigate the baseline brain activity in euthymic BD patients in comparison to healthy controls using a variety of rs-fMRI analyses, such as f/ALFF, ALFF, and ALFF-based FC analysis.

We hypothesize that the aforementioned techniques will differentiate BD and HC, indicating dissimilarities between groups within different brain structures. In addition, we anticipate that each resting-state index will deliver unique data, contributing to a better understanding of the euthymic state.

## 2. Methods

### 2.1. Participants

Forty-two participants were enrolled in this study. The clinical group consisted of 21 euthymic BD patients diagnosed according to DSM-5 and ICD-10 criteria. Euthymia was classified as <11 points in the Montgomery–Asberg Depression Rating Scale (MADRS) [20] and <5 points in the Young Rating Scale for Mania (YMRS) [21]. All the participants were right-handed, as measured by the Neurological Evaluation Scale [22]. Inclusion criterion was treatment with antipsychotic drugs from the group of dibenzoxazepine: clozapine, olanzapine, or quetiapine. Additionally, valproic acid treatment was accepted. Exclusion criteria were: history of alcohol or drug abuse (according to substance use disorder of DSM-5); severe, acute, or chronic neurological and somatic diseases; severe personality disorders; treatment other than mentioned in inclusion criteria.

Healthy controls group consisted of 21 individuals matched in terms of gender and age with BD patients. Exclusion criteria were identical to the ones applying to BD patients. Additionally, participants with a diagnosis of mental illness revealed in the Mini-International Neuropsychiatric Interview or a history of mental illness in first-degree relatives confirmed by the participant during the interview were excluded from the study. All participants signed a written informed consent prior to the assessment. The study was approved by the Jagiellonian University Bioethics Committee (122.6120.125.2015).

### 2.2. MRI Data Acquisition

MRI data were acquired using a 3T Siemens Skyra MR System (Siemens Medical Solutions, Erlangen, Germany). Anatomical images were obtained using sagittal 3D T1-weighted MPRAGE sequence with TR = 2300 and TE = 3.9 ms. A total of 13-minutes functional resting-state (rs-fMRI) BOLD images were acquired using a gradient-echo single-shot echo planar imaging sequence with the following parameters: FOV = 256 mm; TE = 27 ms; TR = 2060 ms; slice thickness = 3 mm; voxel size= 3 × 3 × 3 mm, with no gap. In sum, 39 interleaved transverse slices and 400 volumes were acquired. During the resting state procedure, subjects were instructed to keep their eyes open, to think of nothing particular, and not to fall asleep, which was controlled using an infrared binocular eye tracker (EyeLink 1000 Plus).

### 2.3. fMRI Data Analysis

The rs-fMRI data processing was performed using MATLAB version R2016a (The MathWorks, Inc., Natick, MA, USA) and Data Processing & Analysis for Brain Imaging (DPABI) v. 3.1 [23]. The first 10 time points were discarded due to signal equilibration and then, slice timing was conducted. Many studies have proven that even minor head motions can significantly affect the obtained results [24,25,26,27], therefore head-motion correction was conducted. The subjects with movements in one or more of the orthogonal directions above 3 mm or rotation above 3° were discarded from the analysis; as a result, 2 subjects from BD and 3 subjects from HC were excluded from further analysis. Thus, the final groups consisted of 19 BD patients and 18 HC. Demographic and clinical information of analyzed participants are presented in Table 1. Additionally, the framewise displacement—an index of volume–volume changes in the head position—was evaluated [26]. Then, the voxel-specific framewise displacement (FD_vox_) described by Yan et al. [27] was calculated for each subject and considered as nuisance covariates in the statistical analyses.

Functional images were then linearly normalized to the Montreal Neurological Institute (MNI) space using standard EPI template in SPM 12 (SPM12; Wellcome Trust Centre for Neuroimaging, UCL, London, UK) and spatially resampled to 3 × 3 × 3 mm voxel size. The 24 motion parameters derived from the realignment step, white matter, and cerebrospinal fluid signals were removed by linear regression. According to reports on the risk of detecting false positive values in functional connectivity [28,29], data were not smoothed. Moreover, the global signal was included due to its potential to provide additional valuable information [30]. The signal was then band-pass filtered (0.01–0.08 Hz) to reduce high-frequency noise and low-frequency drift, such as the respiratory and cardiac rhythms.

### 2.4. ALFF and f/ALFF Calculation

As the result of the low timescale of the hemodynamic response, BOLD signal is dominated by the low-frequency fluctuations [31]. Therefore, the amplitude of the low-frequency fluctuations (ALFF) enables to estimate the neural component of the measured BOLD signal, demonstrating how much of the power is in the low-frequency range. While functional connectivity detects similarities in fluctuations of the BOLD signal between two or more regions, ALFF allows to focus on each voxel of the brain, making it a beneficial, complementary method [32]. As for the fractional amplitude of low-frequency fluctuations (f/ALFF), which measures the power within a specific frequency range divided by the total power in the entire detectable frequency range [7], it is found to be more specifically sensitive to neural origins of low-frequency fluctuations [31]. The latter method makes the analysis of low frequency fluctuations even more comprehensive. Both mentioned methods of analysis allow to create a summary map of low-frequency power for each subject, making it beneficial for the study and therefore worth conducting. ALFF along with the f/ALFF were calculated using DPABI v.3.1. [27]. The time series for each voxel was transformed to the frequency domain using a fast Fourier transform. The square root of the power spectrum was calculated, averaged across 0.01–0.08 Hz, and then standardized to z-score by dividing the subject-level maps by the standard deviation. Since ALFF measures the total power of a given time course within a specific frequency range [33], three frequency bands were chosen for the analysis: typical (0.01–0.08 Hz), as well as slow-4 (0.027–0.073 Hz) and slow-5 (0.01–0.027 Hz) [34], which are widely employed in resting-state BOLD fMRI studies [35]. Parallel to ALFF, f/ALFF was calculated in the typical frequency band (0.01–0.08).

### 2.5. ReHo Calculation

Individual regional homogeneity maps were calculated using DPABI v. 3.1 [27]. ReHo analysis depends on Kendall’s coefficient of concordance [36], which was used to measure similarity of the time series of a given voxel and 26 neighbor voxels in voxel-by-voxel manner [37]. Obtained ReHo maps for individual participants were then standardized using Fisher z-transformation in order to perform group analysis. The above method aims to investigate a fundamentally different aspect of resting state brain activity compared with ALFF and f/ALFF and therefore allows to add another potentially valuable marker [31].

### 2.6. Statistical Analyses

For each analysis, FD_vox_ index was included as a nuisance covariate. In order to investigate the differences in typical frequency band of ALFF (0.01–0.08 Hz) between euthymic BD patients and controls, the two-sample *t*-test was performed with Gaussian random field (GRF) correction at voxel *p*-value of 0.01 and cluster *p*-value of 0.05. Then, ALFF-based functional connectivity was calculated where regions displaying differences in ALFF were used as seed points in seed-to-voxel functional connectivity analysis [8]. First level functional connectivity analysis was performed using seed-to-voxel whole brain connectivity at an uncorrected high-threshold of *p* < 0.001. Bivariate Pearson correlation coefficients between the time series of seeds and the rest of the voxels in the brain were extracted and then transformed into Fisher *z*-scores. Group differences were then reported with false discovery rate (FDR) correction at *p*-value of 0.05.

To investigate the main effects of the group, frequency band and their interactions, the two-way ANCOVA was performed using SPM 12 on a voxel-by-voxel basis. The factors were: group (BD and HC) and frequency bands (slow-4 and slow-5) with the inclusion of gray matter mask. The results were reported on voxel-level *p*-value of 0.001, FDR corrected at cluster-level *p*-value of 0.05.

Further post-hoc two-sample *t*-tests were performed for group comparison of the slow-4 and slow-5 bands and the results were reported with GRF correction at voxel *p*-value of 0.01 and cluster *p*-value of 0.05. Furthermore, in order to examine the group differences in f/ALFF, the two-sample *t*-test was performed with GRF correction at voxel *p*-value of 0.01 and cluster *p*-value of 0.05. The cluster exceeding >10 voxels was reported.

Finally, to examine the group differences in ReHo, the two-sample *t*-test was performed with GRF correction at voxel *p*-value of 0.01 and cluster *p*-value of 0.05.

The demographic data were analyzed using SPSS, version 24.0 [38]. Chi squared test was performed for gender distribution and two-sample *t*-test for age and mean head motion.

## 3. Results

There was no significant difference in gender (chi-square χ^2^ = 0.676, df = 1, *p* = 0.614) and age (*p* = 0.42) between the groups (Table 1).

### 3.1. One-Way ANCOVA

Table 2 and Figure 1 showed the one-way ANCOVA analysis of the ALFF values among the groups with FD_vox_ index as a covariate. There was a significant main effect of a group in the left middle frontal gyrus, left insula, right middle occipital gyrus, right gyrus rectus, and left middle temporal pole, as well as a significant main effect of the frequency band (slow-4, slow-5) in the left fusiform gyrus.

### 3.2. Post-Hoc Test

Post-hoc two-sample *t*-test revealed the increased slow-5 ALFF band in the left middle temporal pole within BD in comparison to the HC group (Figure 2, Table 3). There were no significant differences in the slow-4 ALFF band.

### 3.3. Typical ALFF Band: BD vs. HC

Compared to HC, the BD group presented higher ALFF values in the left insula (Figure 3, Table 4). The comparison was conducted using head motion as a covariate.

### 3.4. ALFF-Based FC Analysis

Correlation analyses were conducted using a seed ROI in the left insula. Between-group comparisons revealed significant hyper-connectivity (Table 5, Figure 4) within the BD group in comparison with HC in the bilateral middle frontal gyrus, right superior parietal gyrus, right supramarginal gyrus, left inferior parietal gyrus, left cerebellum, and left supplementary motor area.

### 3.5. fALLF: BD vs. HC

Group differences are shown in Table 6 and Figure 5. Two sample *t*-test showed that compared with HC, the BD group exhibited significantly increased f/ALFF in the left superior temporal gyrus, right putamen, bilateral thalamus, left middle occipital gyrus, and left superior frontal gyrus.

### 3.6. ReHo: BD vs. HC

Compared to HC, the BD group presented higher ReHo values in the left superior medial frontal gyrus and lower ReHo values in the right supplementary motor area (Table 7, Figure 6). The comparison was conducted using head motion as a covariate.

## 4. Discussion

Our results indicate the existence of differences in the spatial patterns of brain activity between euthymic BD patients and HC. To our best knowledge, this is the first study comparing slow-4, slow-5, typical ALFF, and f/ALFF bands in euthymic BD patients and HC. We have shown that each of the techniques indicate dissimilarities between groups within different brain structures. It is important to notice that exact neurophysiological origins ALFF and f/ALFF are unknown, however both measures reflect intensity estimates of spontaneous brain activity [18,39,40,41]. While ALFF represents total power within a specific frequency range, indicating the strength of low-frequency oscillations, f/ALFF demonstrates relative contribution of specific low-frequency oscillations to the whole frequency range. f/ALFF has been described as superior in comparisons to ALFF due to its higher specificity in capturing a gray matter signal [7,18,42]. Our results are consistent with the study of Meda et al. (2015), indicating that f/ALFF is more sensitive in highlighting the differences between BD and controls, capturing more widespread changes compared with ALFF.

The most noticeable differences in our study were revealed in comparison of ALFF/f/ALFF analyses. In the BD patients group, ALFF measurements showed increased activation of the left insula, while f/ALFF indices were increased in the left superior frontal gyrus, left superior temporal gyrus, left middle occipital gyrus, right putamen, and bilateral thalamus. Two independent mayor meta-analyses on neuroimaging studies of bipolar disorder revealed frontal, temporal, occipital as well as some parts of the limbic system regions to be highly associated with many aspects of bipolar disorder [43,44]. We believe that the increased f/ALFF index among numerous fronto-temporo-occipital regions during euthymia might be considered as residual but long-term functional alterations related to the disorder itself. Based on the literature, during depression or manic episodes, those alterations are significantly more evident. Moreover, recently, there has been growing interest in evaluating the role of the insula in BD. The anterior insula cortex has been shown to be involved in emotional processing and regulation as well as cognitive control [45]. Meta-analyses have confirmed a decrease of the anterior insula cortex volume in BD and it has been shown that this region plays a critical role in abnormal mood regulation in this disorder [45,46,47]. Our results correspond with studies indicating aberrant functional connectivity in BD. Li et al. [48] showed increased connectivity between the middle frontal gyrus and insula. It has been proposed that this finding may reflect an extension of compensatory insula hyperactivity due to inefficient dorsolateral prefrontal cortex function which is a part of the middle frontal gyrus [45,49,50]. Altered functional connectivity between the bilateral dorsal anterior insula and left inferior parietal lobule, a key node of the frontoparietal executive control network, has been shown to be differentiated in BD patients with unipolar depression and healthy controls [51,52]. In the study of Sobczak et al. [53], euthymic bipolar disorder patients revealed that the salience network, specifically the insular cortex, might be a key factor in suicidal risk propeness. The study showed that decreased connectivity between regions involved in the salience network has negative correlation with direct suicidal risk. The insular cortex was proven to be involved in, among others, prospective thinking, sensory integration, and many memory-linked processes [54]. Therefore, we hypothesized increased insular cortex activity among euthymic patients to be related to impaired sensory integration as well as processing present experiences [55]. Noteworthy, the aforementioned abilities are thought to be associated with the duration of episodes of depression and mania [56]. In our study, typical band ALFF analysis revealed increased activity in the left insula. This region was chosen as a ROI for seed-to-voxel analysis performed to establish whether a region with abnormal ALFF presents disrupted functional connectivity with other brain areas. Using this approach, we identified network differentiating BD patients from the HC group. We have shown significant hyper-connectivity within the BD group between the left insula and bilateral middle frontal gyrus, right superior parietal gyrus, right supramarginal gyrus, left inferior parietal gyrus, left cerebellum, and left supplementary motor area. 

Our results have also shown increased f/ALFF in the left superior frontal gyrus and right putamen. These findings are consistent with the study of Meda et al. [18] indicating abnormal regional ALFF/f/ALFF activation in fronto-striatal regions in BD patients. The authors suggested that the abnormal modulation of those low frequency oscillations in the aforementioned structures may contribute to abnormal personality and emotional regulation in BD. Disruptions of f/ALFF in the frontal gyrus and putamen stay in line with the well-described fronto-striatal loop dysfunctions in BD [18,57,58,59,60].

The subdivision of ALFF into slow-4 and slow-5 bands allowed to unveil further between-group differences. We have shown that BD patients reveal increased slow-5 ALFF in the left middle temporal pole. This result is consistent with the observation of increased slow-5 within the interior/middle temporal gyrus of psychotic BD patients [18] and common alteration of slow-5 in the left temporal gyrus in BD, schizophrenia, and major depressive disorder [61]. It has been shown that BD patients present disrupted functional connectivity between the left middle temporal gyrus and Heschl’s gyrus which may be associated with language-related symptoms in that disorder [62]. The study of Li, Xu, and Lu [48] confirmed the above alterations associated with affective disorders by revealing increased ReHo in the middle temporal gyrus. The aforementioned results suggest some functional impairments might be present even during the remission and subjective well-being. 

None of the previous studies performed whole-brain ReHo analysis of euthymic BD patients. Whereas, the current study showed increased ReHo values in the left superior medial frontal gyrus and decreased values in the right supplementary motor area. Recently, Wang et al. [63] showed that BD patients presented significant rightward hemispheric asymmetry only in the supplementary motor area. Other studies revealed that during motor task, s patients present greater activity in the right supplementary motor area [64,65] than those in the left. It has been proposed that this brain region may play the role of a potential neural marker of altered asymmetries in nodal efficiency for clinical presentation of BD. Our observation of increased ReHo in the left superior medial frontal gyrus stays in line with results of Liu et al. (2012) indicating higher ReHo values in the left medial frontal gyrus in depressed BD patients. It has been proposed that the overactivity of this region may be responsible for the cognitive–emotional interference seen in BD (Liu et al., 2012). Abnormal ReHo of this region has been also speculated to be a candidate for either trait or state marker related to BD depressive episode (Liu et al., 2012). Our results showed increased ReHo within the left medial frontal gyrus, indicating this abnormality may be state-independent, however further studies are required to verify those observations.

We are aware of several limitation of this study such as: (a) relatively small number of participants; (b) BD patients’ heterogeneity—group consisted of BD I and BD II patients, as well as patients with a history of psychotic symptoms; (c) the BD patients group was not drug naive, relatively small number of subjects makes it impossible to control the effect of medication.

## 5. Conclusions

We believe that this is the first rs-fMRI study combining ReHo, ALFF, f/ALFF, and subdivided frequency bands (slow-4 and slow-5) in euthymic BD patients. ReHo, ALFF, f/ALFF, and slow-5 analysis revealed significant differences between the two studied groups. However, those four methods indicated various brain structures, without a single region overlapping between them. This indicates that a combination of rs-fMRI analyses methods may complement each other, revealing complex resting state abnormalities in BD.

## Figures and Tables

**Figure 1 brainsci-11-00599-f001:**
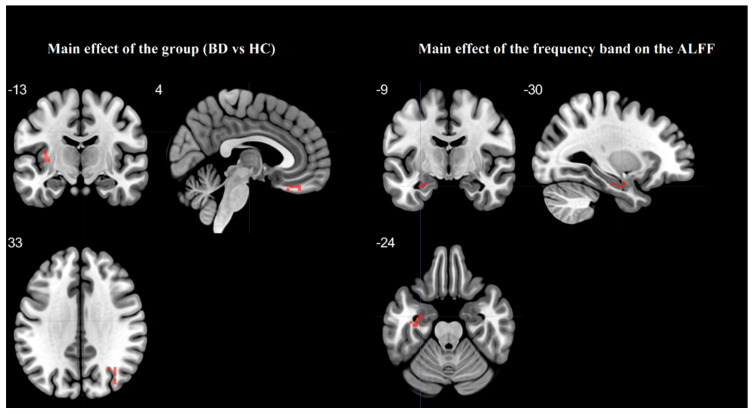
The main effect for the group on ALFF and the main effect of the frequency band on the ALFF with differences between the slow-4 and slow-5, based on one-way ANCOVA.

**Figure 2 brainsci-11-00599-f002:**
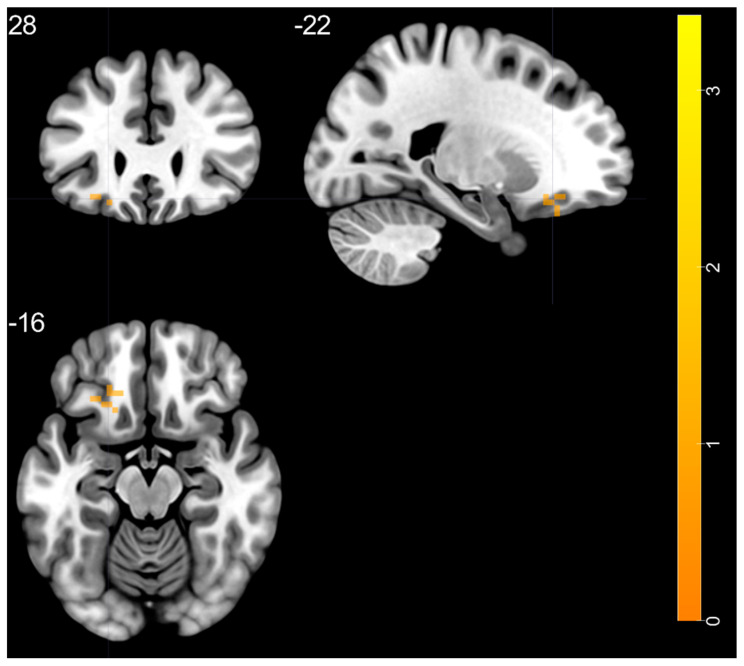
Post-hoc two-sample *t*-test with increased slow-5 oscillations ratio within BD in comparison to the HC group, BD—bipolar disorder, HC—healthy control.

**Figure 3 brainsci-11-00599-f003:**
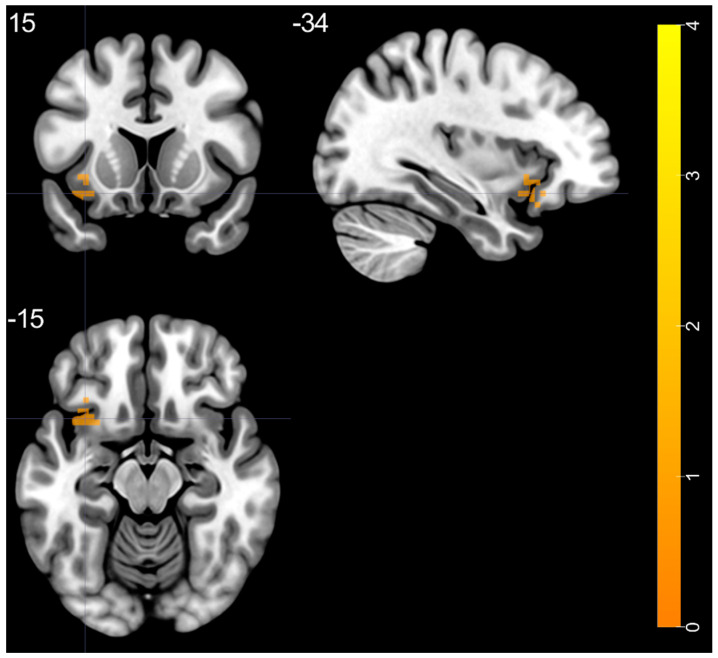
Increased values in the typical ALFF band (0.01 Hz–0.08 Hz) in the BD compared to HC group, BD—bipolar disorder, HC—healthy control.

**Figure 4 brainsci-11-00599-f004:**
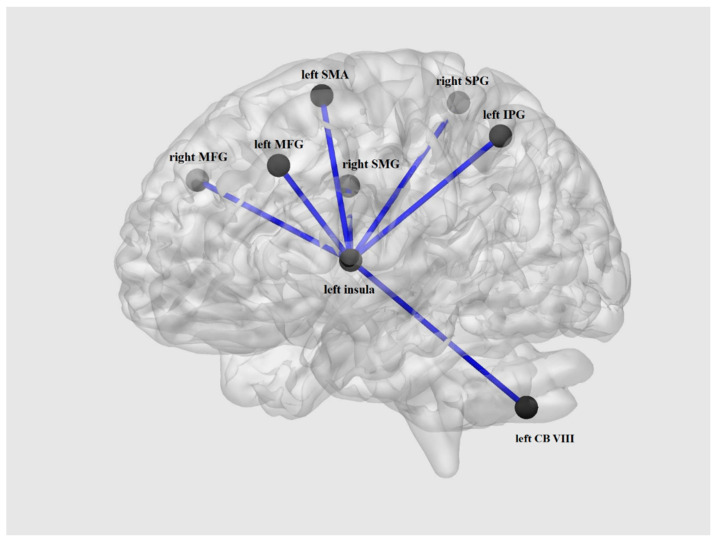
Between group comparisons of seed-to-voxel connectivity revealing hyper-connectivity within the BD group. MFG—middle frontal gyrus, SPG—superior parietal gyrus, SMG—supramarginal gyrus, IPG—inferior parietal gyrus, CB—cerebellum, SMA—supplementary motor area, BD—bipolar disorder.

**Figure 5 brainsci-11-00599-f005:**
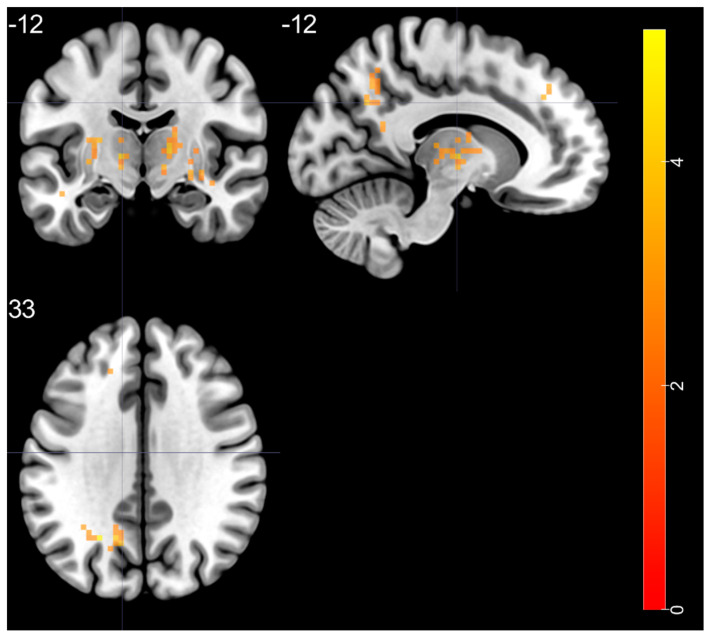
Group differences in f/ALLF, obtained by two sample *t*-test. Analysis show increased f/ALFF within the BD group, BD—bipolar disorder.

**Figure 6 brainsci-11-00599-f006:**
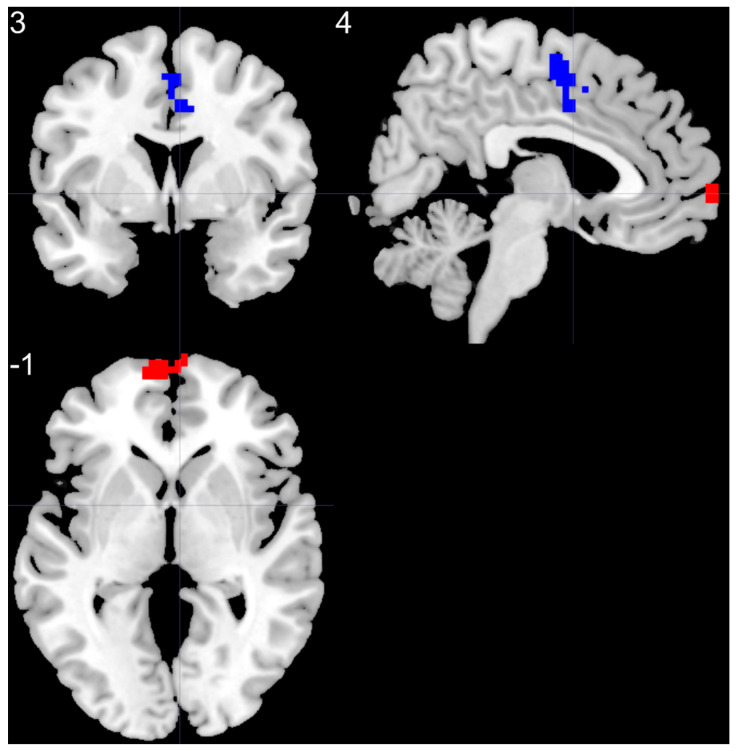
Group differences in ReHo analysis, obtained by two sample *t*-test. BD patients presented higher ReHo values in the left superior medial frontal gyrus (red color) and lower ReHo values in the right supplementary motor area (blue color), BD—bipolar disorder.

**Table 1 brainsci-11-00599-t001:** The description of study groups. BD—bipolar disorder, HC—healthy controls, SD–standard deviation. Head motions were evaluated according to the frame-wise displacement criteria by Van Dijk et al. (2012).

	BD Group	HC Group	*p*-Values
Age (years, mean (SD)) ^a^	36 (6.4) 7/12 9/10	35 (10.2)	0.420
Sex (men/women) ^b^		9/9	0.411
BD type (I/II)		-	
Number of BD patients with history of psychotic symptoms	5	-	
Duration of treatment (years, mean (SD))	6.6 (6.1)		
Number of affective episodes (mean, (SD))	9.6 (11.0)		
Number of hypomanic episodes (mean, (SD))	1.1 (1.4)		
Number of manic episodes (mean, (SD))	2.7 (5.7)		
Number of depressive episodes (mean, (SD))	5.8 (6.5)		
Mean head motion ^a^	0.077 (0.07)	0.073 (0.077)	0.003
Medication			
	Number of patients (%)	Dose (mean mg (SD))	
Quetiapine	6 (32%)	367.7 (233.8)	
Olanzapine	7 (37%)	9.6 (4.7)	
Valproic acid	10 (53%)	980 (315.5)	

^a^ *t*-test, ^b^ Chi-square test.

**Table 2 brainsci-11-00599-t002:** Significant main effects of the group (BD vs. HC) and frequency band (slow-4 and slow-5) by one-way ANCOVA.

Brain Regions	F-Scores	MNI Coordinates			Cluster Size (Voxels)
		x	y	z	
Main effect of group					
left MFG—BA 10	37.88	−30	63	3	15
left insula—BA 47	26.35	−36	15	−15	10
right MOG—BA 19	23.22	30	−63	36	16
right RG—BA 11	21.72	6	33	−24	13
left MTP—BA 21	20.38	−39	15	−42	14
Main effect of frequency band					
left FG—BA 37	17.20	−36	−18	−24	11

Note: MFG—middle frontal gyrus, MOG—middle occipital gyrus, RG—gyrus rectus, MTP—middle temporal pole, FG—fusiform gyrus, BD—bipolar disorder, HC—healthy control; BA—Brodmann area; MNI—Montreal Neurological Institute. FDR (false discovery rate) corrected on cluster-level *p* < 0.05, a minimum cluster size of 10 voxels.

**Table 3 brainsci-11-00599-t003:** Results of post-hoc tests (two-sample *t*-test).

Brain Regions	Peak T-Scores	MNI Coordinates			Cluster Size (Voxels)
		x	y	z	
In the slow-4 band					
BD < HC					
None					
BD > HC					
None					
In the slow-5 band					
BD > HC					
left MTP—BA 21	3.425	−39	15	−42	41
BD < HC					
None					

Note: MTP—middle temporal pole, BD—bipolar disorder, HC—healthy control; BA—Brodmann area; MNI—Montreal Neurological Institute. GRF (Gaussian random field) corrected with a voxel-level *p* < 0.01 and cluster-level *p* < 0.05, minimum cluster size > 26 voxels.

**Table 4 brainsci-11-00599-t004:** Significant differences in ALFF between BD and HC groups (two-sample *t*-test).

Brain Regions	Peak T-Scores	MNI Coordinates			Cluster Size (Voxels)
		x	y	z	
BD > HC					
left insula—BA 47	4.228	−36	15	−15	37

Note: ALFF—amplitude of low-frequency fluctuation, BD—bipolar disorder, HC—healthy control; BA—Brodmann area; MNI—Montreal Neurological Institute. GRF (Gaussian random field) corrected with a voxel-level *p* < 0.01 and cluster-level *p* < 0.05, minimum cluster size > 26 voxels.

**Table 5 brainsci-11-00599-t005:** Results from seed-to-voxel analysis.

Brain Regions HC	Peak T-Scores	MNI Coordinates			Cluster Size (Voxels)
		x	y	z	
left MFG—BA 46	−5.174	−36	48	30	11
right MFG—BA 9	−5.597	39	36	39	18
right SPG—BA 7	−6.133	30	−63	51	55
right SMG—BA 40	−5.191	57	−42	42	20
left IPG—BA 40	−6.024	−30	−69	42	26
left CB—lobule VIII	−6.114	−12	−72	−51	10
left SMA—BA 6	−5.010	−3	−3	60	10

Note: MFG—middle frontal gyrus, SPG—superior parietal gyrus, SMG—supramarginal gyrus, IPG—inferior parietal gyrus, CB—cerebellum, SMA—supplementary motor area; BD—bipolar disorder, HC—healthy control; BA—Brodmann area; MNI—Montreal Neurological Institute; FDR (false discovery rate) corrected cluster-level *p* < 0.05; minimum cluster size > 10 voxels.

**Table 6 brainsci-11-00599-t006:** Significant differences in f/ALFF based on the main effects of disease between BD and HC groups (two-sample *t*-test).

Brain Regions	Peak T-Scores	MNI Coordinates			Cluster Size (Voxels)
		x	y	z	
BD > HC					
left STG—BA 20	4.525	−48	−24	−6	38
right putamen—BA 48	5.178	33	−6	0	144
right thalamus (ventral lateral nucleus)	4.28	15	−12	6	67
left thalamus (ventral lateral nucleus)	4.351	−12	−12	3	113
left MOG—BA 19	5.075	−24	−60	33	58
left SFG—BA 9	3.741	−9	42	39	27

Note: f/ALFF—fractional amplitude of low-frequency fluctuation, STG—superior temporal gyrus, SFG—superior frontal gyrus, MOG—middle occipital gyrus, BD—bipolar disorder, HC—healthy control; GRF (Gaussian random field) corrected with a voxel-level *p* < 0.01 and cluster-level *p* < 0.05, minimum cluster size > 26 voxels.

**Table 7 brainsci-11-00599-t007:** Results of two-sample *t*-test from ReHo analysis comparing BD and HC groups.

Brain Regions	Peak T-Scores	MNI Coordinates			Cluster Size (Voxels)
		x	y	z	
BD > HC					
left SMF	5.37	−6	66	0	85
BD < HC					
right SMA	−4.14	3	−6	57	70

Note: SMF—superior medial frontal gyrus, SMA—supplementary motor area, BD—bipolar disorder, HC—healthy control, BA—Brodmann area, MNI—Montreal Neurological Institute; GRF (Gaussian random field) corrected with a voxel-level *p* < 0.01 and cluster-level *p* < 0.05, minimum cluster size > 26 voxels.

## Data Availability

Data is available on request.

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
