# Peer review of "Time-Frequency Characterization of Resting Brain in Bipolar Disorder during Euthymia—A Preliminary Study"

_brainsci, 2021, doi:10.3390/brainsci11050599_

Round 1

Reviewer 1 Report

  1. How can task-negative states be useful on a molecular level?
  2. Is there any euthymic state that a 'healthy control' will encounter?
  3. What is the test-retest reliability estimate for a small-world network relative to a more complex network?

Figure 2 is confusing as the BD and HC aren't clearly labeled

Author Response

How can task-negative states be useful on a molecular level?

Thank you for that valuable comment.

As stated in Bartelle, Barandov & Jasanoff, (2016), the molecular fMRI techniques have particular potential to combine the specificity of cellular-level measurements with the noninvasive whole-brain coverage of fMRI. In our work on the other hand, we were focused on more global level by using techniques like ALFF, fALFF and REHO which are whole brain techniques.

Bartelle, B. B., Barandov, A., & Jasanoff, A. (2016). Molecular fMRI. Journal of Neuroscience, 36(15), 4139-4148,

Is there any euthymic state that a 'healthy control' will encounter?

 No, in our article we refer to euthymia as the state of relative mood balance, without the symptoms of hypomania, mania, or depression.

What is the test-retest reliability estimate for a small-world network relative to a more complex network?

According to the study of Holiga et al. (2018) test-retest reliability and reproducibility performance of ALFF, f/ALFF and ReHo ranged from good to excellent. Below are the Intraclass correlation coefficients for each index between 2 sessions:

ALFF: Between subject reliability: 0.72, Within subject reliability: 0.96,

Whole brain voxel-wise reliability: 0.55

fALFF: Between subject reliability: 0.57, Within subject reliability: 0.98,

Whole brain voxel-wise reliability: 0.37

ReHo: Between subject reliability: 0.58, Within subject reliability: 0.96,

Whole brain voxel-wise reliability: 0.46

Holiga, Š., Sambataro, F., Luzy, C., Greig, G., Sarkar, N., Renken, R. J., Dukart, J. (2018). Test-retest reliability of task-based and resting-state blood oxygen level dependence and cerebral blood flow measures. PloS one, 13(11), e0206583.

Reviewer 2 Report

Title: Time-frequency characterization of resting brain in Bipolar Disorder during Euthymia - a preliminary study

General

The manuscript titled “Time-frequency characterization of resting brain in Bipolar Disorder during Euthymia - a preliminary study” investigated the baseline brain activity in euthymic bipolar disorder (BD) patients by comparing it to healthy controls (HC) with the use of variety of resting state functional magnetic resonance imaging (rs-fMRI) analyses, such as Amplitude of Low Frequency Fluctuations (ALFF), fractional ALFF (f/ALFF), ALFF-based functional connectivity (FC) and Regional Homogeneity (ReHo). The results obtained are well furnished and in this rs-fMRI study, the authors combined ReHo, ALFF, f/ALFF and subdivided frequency bands (slow-4 and slow-5) in euthymic BD patients. ALFF, f/ALFF, slow-5 as well REHO analysis revealed significant differences between two studied groups. This is an interesting study that does advance knowledge; however, some points merit further consideration.

Major compulsory revisions

Abstract:

- Results are very vague.

- There is no statically significance in the abstract section.

- Conclusion is very vague.

Introduction:

- The authors investigated on the Bipolar Disorder during euthymia and advancements in the field of functional magnetic resonance imaging (fMRI) methods, without any scientific judgment at these points, I decided that it cannot be accepted because the Bipolar Disorder and the previous methods time-frequency characterization of resting brain are not properly explained and documented. A proper study or documentation supporting that, including the scientific confirmation of previous used techniques, limitations and their specifications are needed.

“The goal of this paper is to investigate the baseline brain activity in euthymic BD patients in comparison to healthy controls using a variety of rs-fMRI analyses, such as f/ALFF, ALFF and ALFF-based FC analysis”.   This is overall objective or specific objectives?

- Problem statement, hypothesis, and overall and specific objectives of this research should be included in the introduction section.

Material and methods:

- How many participants used for studies and? Male and female? Add the sample size (n per sex). Was this number accepted by the Ethical Committee? Would be interesting to include in the text the number of the document sent by the ethics committee (Institutional Ethical Clearance Number). 

- The manufacturer of the devices used must be provided.

- Without citing a reference for the used sample size, justification must be provided for studying only 42 participants in the method section.

Results

Table 1: a) t-test; b) Chi-square test. Why using two different statistical analyses?

In figure 1: Please, improve your figures with high resolution. Additionally, indicate with narrows each finding to understand it. Idem for figure 2, figure 3, figure 5, figure 6

Discussions

- I am of the view that the discussion section of this article should be treated with more details and consistency. Readers might expect your discussion to be extended to possible explanations or justifications of findings. Based on previous knowledge, postulates can be posed in case no scientific arguments are available to explain the results obtained. It is always important to compare your results with previous findings and refer to similar works in the field.

Conclusions

- Conclusion is very vague.

Level of Interest

- Check for grammatical mistakes throughout the text according to the instruction for authors.

- An article of importance in its field.

Author Response

Results in the abstract are very vague.

In the results section, we reported significant results or the lack of it from all the methods that we have used.

There is no statically significance in the abstract section.

Than you for that notion. We did not put the statistical significance there because of the word limitation in the abstract section.  

- Conclusion is very vague.

We understand. We tried to make that section clearer. We wanted to point out the fact that the results from different methods did not overlap which might be proof that our results are complementary to each other and that all of these various techniques should be analyzed together in order to see a bigger perspective. The changes were made in lines: 288-295, 307-315, 323-327, 337-340.

Introduction:

- The authors investigated on the Bipolar Disorder during euthymia and advancements in the field of functional magnetic resonance imaging (fMRI) methods, without any scientific judgment at these points, I decided that it cannot be accepted because the Bipolar Disorder and the previous methods time-frequency characterization of resting brain are not properly explained and documented. A proper study or documentation supporting that, including the scientific confirmation of previous used techniques, limitations and their specifications are needed.

Thank you for that suggestion. In fact, previously the different specific bands for ALFF as well as fALFF were used in order to characterize various psychiatric and neurological diseases such as:

MCI: Zhang, T., Zhao, Z., Zhang, C., Zhang, J., Jin, Z., & Li, L. (2019). Classification of early and late mild cognitive impairment using functional brain network of resting-state fMRI. Frontiers in psychiatry, 10, 572.

Depression: Wang, L., Kong, Q., Li, K., Su, Y., Zeng, Y., Zhang, Q., ... Si, T. (2016). Frequency-dependent changes in amplitude of low-frequency oscillations in depression: a resting-state fMRI study. Neuroscience letters, 614, 105-111.

Schizophrenia: Luo, Y., He, H., Duan, M., Huang, H., Hu, Z., Wang, H., ... Luo, C. (2020). Dynamic functional connectivity strength within different frequency-band in schizophrenia. Frontiers in psychiatry, 10, 995.

Our work is the first one depicting diffrences in AlFF, fALFF and ReHo in euthymic patients.

“The goal of this paper is to investigate the baseline brain activity in euthymic BD patients in comparison to healthy controls using a variety of rs-fMRI analyses, such as f/ALFF, ALFF and ALFF-based FC analysis”.   This is overall objective or specific objectives?

Thank you for that comment. The objective you are referring to is an overall objective. The specific objectives were added in lines: 114-121.

- Problem statement, hypothesis, and overall and specific objectives of this research should be included in the introduction section.

The specific objectives were added in lines: 114-121.

Material and methods:

- How many participants used for studies and? Male and female? Add the sample size (n per sex). Was this number accepted by the Ethical Committee? Would be interesting to include in the text the number of the document sent by the ethics committee (Institutional Ethical Clearance Number). 

In the group diagnosed with bipolar disorder were 7 men and 2 women while in control group it was 9 men and nine women. The Ethical Committee approval number was added in line 141.

- The manufacturer of the devices used must be provided.

This information can be found in line150.

- Without citing a reference for the used sample size, justification must be provided for studying only 42 participants in the method section.

As we mentioned in our title and in our manuscript, our study is a preliminary work. In the future studies we want to expand the sample size. Previously the ALFF, fALFF and ReHO indexes were used on similar sample size in order to characterize different psychiatric disorders like depression and schizophrenia:

Wang, L., Dai, W., Su, Y., Wang, G., Tan, Y., Jin, Z., ... & Si, T. (2012). Amplitude of low-frequency oscillations in first-episode, treatment-naive patients with major depressive disorder: a resting-state functional MRI study. PloS one7(10), e48658.

Song, Y., Shen, X., Mu, X., Mao, N., & Wang, B. (2020). A study on BOLD fMRI of the brain basic activities of MDD and the first-degree relatives. International journal of psychiatry in clinical practice24(3), 236-244.

Results

Table 1: a) t-test; b) Chi-square test. Why using two different statistical analyses?

We used t-test and Chi-square for different reasons. T-test is a parametric test which is usually used for comparing two groups when the variable is quantitative and all the assumptions are met, like the age variable. Chi square is a nonparametric test which allowed us to check the equipotency of the group according to a gender, which is a qualitative.

In figure 1: Please, improve your figures with high resolution. Additionally, indicate with narrows each finding to understand it. Idem for figure 2, figure 3, figure 5, figure 6

Thank you for that notion. Higher resolution figures will be added at the final step during the author’s proof.

Discussions

- I am of the view that the discussion section of this article should be treated with more details and consistency. Readers might expect your discussion to be extended to possible explanations or justifications of findings. Based on previous knowledge, postulates can be posed in case no scientific arguments are available to explain the results obtained. It is always important to compare your results with previous findings and refer to similar works in the field.

Thank you for that notion. The discussion and conclusions were updated in accordance with your suggestions.

Round 2

Reviewer 2 Report

Title: Time-frequency characterization of resting brain in Bipolar 1 Disorder during Euthymia-a preliminary study

Manuscript ID: brainsci-1180649

Version: Date 30 April 2021

Reviewer’s report: Authors have addressed almost all the issue I raised out from the initial manuscript.

Major compulsory revisions: None

Minor essential revisions: None

Discretionary Revisions: None

Level of interest: An article of importance in its field

Quality of written English: Acceptable

Declaration of competing interests: I have no competing/conflicting interest whatsoever.